# any4: Learned 4-bit Numeric Representation for LLMs

**Mostafa Elhoushi** [* 1]   **Jeff Johnson** [* 1]

## Abstract

We present any4, a learned 4-bit weight quantization solution for large language models (LLMs) providing arbitrary numeric representations without requiring pre-processing of weights or activations. any4 yields higher accuracy compared to other related 4-bit numeric representation types: int4, fp4 and nf4, as evaluated on a range of model sizes, generations and families (Llama 2, Llama 3, Mistral and Mixtral). While any4 does not require preprocessing of weights or activations, it is also competitive with orthogonal techniques that require such preprocessing (e.g., AWQ and GPTQ). We also experiment with any3 and any2 and show competitiveness at lower bits. Additionally, we show that we can calibrate using a single curated diverse sample rather than hundreds of samples from a dataset as done in most quantization approaches. We also open source tinygemm, a latency optimized GPU matrix multiplication library for LLMs, that implements any4 using a GPU-efficient lookup table strategy along with other common quantization methods. We open source our code at https://github.com/facebookresearch/any4.

## 1. Introduction

Reduced neural network parameter sizes are important for efficient inference, whether at datacenter scale, where accelerators can be provisioned based more upon arithmetic throughput rather than memory requirements, or with edge devices, where smaller, slower memories could be used improving battery lifetime while meeting performance constraints. Given training is typically done in high dynamic range floating point arithmetic, techniques to lossily compress weights must deal with the possibility of varying scale factors and outliers. Various weight numeric formats, such

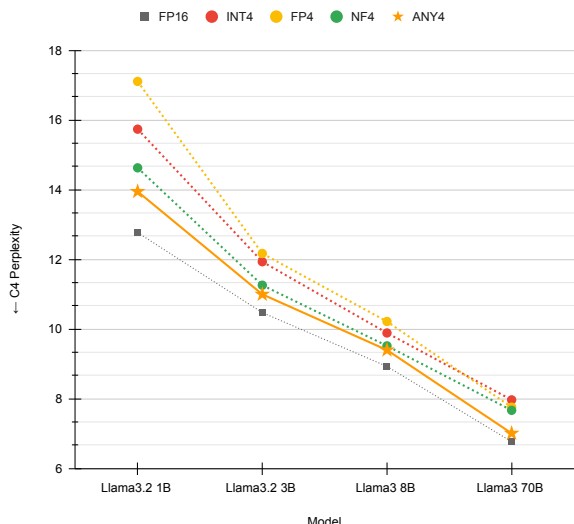

Figure 1: Perplexity by quantizing various Llama3 model sizes. Our proposed any4 is the most accurate across numeric formats.

as 4-bit integer (int4), floating point (fp4), or custom distributions such as NormalFloat4 (nf4) (Dettmers et al., 2023)) along with quantization grouping (Dai et al., 2021) are used to increase accuracy. Pre-processing weights and/or activations (e.g., AWQ (Lin et al., 2024), GPTQ (Frantar et al., 2023), or weight Hadamard transforms (Ashkboos et al., 2024b; Liu et al., 2024) can aid with accuracy as well. In this paper, we present a new learned numeric representation, any4, that does not require online or offline modification of weights or activations. any4 quantization accuracy outperforms other numeric representation types, and is competitive with orthogonal quantization algorithms that preprocess weights and/or activations (orthogonality implying that some of these techniques can be applied together with any4 representation). Accuracy was evaluated on a wide range of model sizes, generations and families.

## 2. Background

Trained neural network weights tend to be roughly Gaussian in nature but with heavier tails (Goodfellow et al., 2016). In attempting to lossily compress weights via quantization (yielding fewer reproduction values than the original do-

---

[*]Equal contribution   [1]FAIR at Meta.   Correspondence to: Mostafa Elhoushi <m.elhoushi@ieee.org>, Jeff Johnson <jhj@meta.com>.

*Proceedings of the 42nd International Conference on Machine Learning*, Vancouver, Canada. PMLR 267, 2025. Copyright 2025 by the author(s).

main), being able to closely match the weight distribution with post-quantization possible reproduction values is important for accuracy.

## 2.1. Uniform Integer Quantization

Some of the first neural network quantization works concerned uniform integer quantization (Jacob et al., 2018). Given a set of values to quantize, we obtain the maximum absolute value, and set that to the extreme value (e.g., -128 / +127 for int8 and -8 / +7 for int4 quantization), with zero being preserved (int8/int4 zero dequantizes to original domain zero). Each increment between int8/int4 values corresponds to a fixed increment (scale) in the original floating point domain.

This allows for more efficient (chip area and power) hardware circuits, as integer multiply-add is much simpler than floating point multiply-add. However, uniform integer quantization is best suited to representing samples from a uniform distribution, which is a mismatch with neural network properties. Increased bitwidth (more dense uniform samples) is needed for accuracy due to the expected distribution mismatch, indicating that there is waste in memory storage.

## 2.2. Floating Point Quantization

Floating point quantization (reducing fractional precision and dynamic range via rounding) is another mechanism. Unlike integer quantization, reproduction values are now non-uniformly spaced. Floating point arithmetic is a piecewise linear distribution of values: the steps between floating point exponents are geometric in nature (multiply or divide by 2 each increment), but within a given exponent value, the spacing of reproduction values is linear (as given by the significand bits). This is slightly closer mapping as a Gaussian distribution with zero mean has most of the mass of the distribution at smaller exponent values more densely sampled by floating point than linear distributions on the number line, while within an exponent the spacing of values is still linear.

Such quantization makes sense with hardware support for reduced bit width floating point types (e.g., fp8 formats with Nvidia's H100 GPU and fp4 with Nvidia's B100 GPU). In lieu of native conversion instructions, bit manipulation can usually convert or round a $n$-bit fp$n$ value to the nearest standard fp16/bf16 value (thus, fp4 can be emulated on devices with higher bit width floating point support).

## 2.3. Grouped Quantization

As the bitwidth (and thus the number of possible quantization reproduction values) decreases, it can be useful to introduce metadata pertaining to groups of values to the quantization to improve accuracy, with metadata storage cost amortized across many values (Darvish Rouhani et al., 2020). Grouped quantization is an attempt at this. Instead of forcing a single scalar value itself to be the entire representation, we can define groups of contiguous values along a row or column of the matrix. A common offset and scale factor is defined for a group of values such that the reconstruction error is improved, with typical group sizes in practice being 32 - 256. Other variants include Shared Microexponents (Rouhani et al., 2023), providing a group-wise shared exponent value (multiplicative scale) to adjust per-scalar 4 bit floating point values (MX4) in lieu of a scale and offset.

## 2.4. Non-Uniform Quantization

Thus far we have discussed uniform (linear) and floating-point (log/linear) distributions. But we can go further and have quantization reproduction values match the seen distributions more closely.

**NormalFloat4** (**nf4**) (Dettmers et al., 2023) attempts to do exactly this by having the reproduction values (fixed ahead of time) match a Gaussian distribution exactly. However, with an even number of reproduction values (e.g., $2^n$ for $n$ bits), we cannot represent a Gaussian symmetrically if we wish to preserve zero. So nf4 is asymmetric, using one of the 16 values to represent zero. This results in higher accuracy, especially for partially sparse matrices.

**AbnormalFloat4** (**af4**) (Yoshida, 2023) is a variant of nf4 which adjusts the distribution based on quantization group size. The larger the quantization group, the larger the expected maximum absolute value of Gaussian distribution samples, but the mass of the distribution would still be close to 0. Mapping the nf4 distribution based on the seen absolute maximum value would result in much of the mass of the distribution (values closer to the mean) not being as accurately represented. af4 adjusts the distribution based on group size to take this into account.

### 2.4.1. Arbitrary Non-uniform Quantization: any4

Instead of trying to match an a priori data distribution as nf4/af4 do, we can instead learn the distribution from the seen data itself. This was explored in signal processing (Lloyd, 1982a; Max, 1960) and any4 explores this for LLMs. For each set of values along each row of a matrix, we can perform k-means (Lloyd, 1982b; MacQueen et al., 1967) or neural network-based clustering, so each row of the matrix has its own 4-bit quantization code, providing indices into a per-row codebook or lookup table (LUT) containing arbitrary floating point dequantization values. This adds little overhead to quantization: for each row of a M×4096 matrix, any4 will add 16 bfloat16/float16 values, for an overhead of (16 × sizeof([b]float16) × 8 bits/byte) / 4096 columns = 0.0625 bits for each matrix entry. Like existing

4-bit techniques, for higher accuracy we add quantization groups (e.g., each set of $g$ contiguous row values has a shared 16-bit scale and zero point). Thus, per-scalar quantization group overhead for $g$ = 128 in our example would be ((4096 / 128) × (2 × 16)) / 4096 = 0.25 bits, yielding 0.0625 + 0.25 + 4 = 4.3125 bits for any4 representation. Note that standard int4 grouped quantization is already 4.25 bits/entry here, with extension to any4 only adding 0.0625 bits/entry of LUT overhead.

In addition, the likely most efficient way to implement nf4 and af4 in software itself is via the same mechanism as any4: using a LUT, as there is no efficient programmatic way to convert a 4-bit integer to an nf4/af4 value using a small number of instructions. To support nf4/af4, our CUDA implementation also allows using a single 16 entry any4 LUT for an entire matrix instead of a LUT per each matrix row. This paper solely evaluates the latter.

## 2.5. Quantization Process

Vanilla quantization happens in 2 steps: scaling followed by rounding.

### 2.5.1. SCALING

Numeric formats have different numeric ranges, and high precision numeric formats usually have orders of magnitude larger ranges from low precision numeric formats, e.g., fp32 ranges from $-3.4 \times 10^{38}$ to $+3.4 \times 10^{38}$ while int4 ranges from -7 to +8. Moreover, the numeric range of a given tensor could be orders of magnitude different from a low precision format (e.g., most weight values range from -0.01 to +0.01 while int4 ranges from -7 to +8). Hence, directly rounding each element in a tensor to its nearest value in a numeric format will waste most of the bits and lead to high reconstruction error.

Instead, most approaches scale a tensor, or a subset of a tensor, to the range of lower precision numeric format. Given a weight tensor $w$, and an index $i$, the scaled weight tensor, $w_S$, can be expressed as:

$$w_{S_i} = \frac{w_i - \beta_i}{\alpha_i} \tag{1}$$

Scale factors $\alpha$ and $\beta$, are high precision scalar values that are calculated for each group of indices, $G$. For asymmetric quantization[1]:

$$\alpha_{j \in G} = \frac{\max(w_{j \in G}) - \min(w_{j \in G})}{Q_{max} - Q_{min}} \tag{2}$$
$$\beta_{j \in G} = \min(w_{j \in G})$$

[1]Note that some quantization literature scale slightly differently from us: $\beta_{j \in G} = \text{round}(\min(w_{j \in G})/\alpha_{j \in G})$ and $w_{S_i} = \frac{w_i}{\alpha_i} - \beta_i$

For symmetric quantization:

$$\alpha_{j \in G} = \frac{\max(\text{abs}(w_{j \in G}))}{Q_{max}} \tag{3}$$
$$\beta = 0$$

where $G$ is a set of indices of a tensor, $\alpha$ and $\beta$ are scaling factors, $Q_{min}$ and $Q_{max}$ are the minimum and maximum values of the lower precision numeric format.

Scaling could be applied at different granularities:

- **Tensorwise:** where $G$ is the set of all indices of the tensor. Hence, all elements in tensor, $w$, share the same scale factors: $\alpha_{i,j} = \alpha, \beta_{i,j} = \beta, \forall i, j$.

- **Rowwise:** where $G$ is the set of all indices of a row. Elements in each row of a tensor share the same scale factors: $\alpha_{i,j} = \alpha_i, \beta_{i,j} = \beta_i, \forall j$.

- **Columnwise:** where $G$ is the set of all indices of a column. Elements in each column of a tensor share the same scale factors: $\alpha_{i,j} = \alpha_j, \beta_{i,j} = \beta_j, \forall i$.

- **Groupwise:** where $G$ is the set of non-overlapping consecutive indices along a row (or column), of size $1 \times g$, where group size, $g$, is a scalar hyperparameter. Elements in each group, $G_k$, share the same scale factors: $\alpha_{i,j} = \alpha_{i,G_k}, \beta_{i,j} = \beta_{i,G_k}, \forall j$ s.t. $kg \leq j < k(g+1)$. Values of 64 or 128 for $g$ usually provide a sweet spot between accuracy and overhead for 4-bit quantization.

- **Blockwise:** where $G$ is the set of indices within a two-dimensional block of size $b \times b$, where, $b$, is a scalar hyperparameter. Elements in each block, $G_{k,l}$, of a tensor share the same scale factors: $\alpha_{i,j} = \alpha_{G_{k,l}}, \beta_{i,j} = \beta_{G_{k,l}}, \forall i, j$ s.t. $kb \leq i < k(b+1), lb \leq j < l(b+1)$.

In our work, we focus on weight-only groupwise quantization (along the reduction dimension) and, unless stated otherwise, use a default group size $g$ of 128.

### 2.5.2. ROUNDING

After scaling, the next step is to round the scaled value to the nearest value in the low-precision quantization format:

$$w_Q = \text{round}_Q(w_S) \tag{4}$$

And to dequantize: $\text{dequant}(w_Q) = \alpha w_Q + \beta$.

## 3. Related Work

Quantization has long been researched to run on CPUs and custom chips (Xie & Jabri, 1992). Various techniques can be categorized into:

**Weights vs. Activations vs. Gradients vs. Optimizer States** Quantization can be applied on weights only (AWQ (Lin et al., 2024), GPTQ (Frantar et al., 2023)), weights and activations (SmoothQuant (Xiao et al., 2023), LLM.int8() (Dettmers et al., 2022a)), KV cache (KVQuant (Hooper et al., 2024)), and can be applied to gradients for training (TinyScript (Fu et al., 2020)) and optimization states (8-bit Optimizers (Dettmers et al., 2022b)). Auto-regressive decoding with batch size 1 and sequence length 1 is a highly memory bound process (a big portion of compute time is spent in loading weights compared to processing activations), thus 4-bit weight only quantization leads to better speedup than 8-bit weight and 8-bit activation quantization (PyTorch, 2024). Moreover, 4-bit weight only quantization leads to a better accuracy-speed tradeoff compared to 4-bit weight and 4-bit activation quantization. In this research, we focus on quantizing weights only.

**Post-Training Quantization (PTQ) vs. Quantization Aware Training (QAT)** PTQ refers to quantization on a trained model without the need for further training. QAT refers to quantization during training, whether training a model from scratch, e.g., FP8-LM (Peng et al., 2023), or continually training or finetuning a trained model, e.g., QLoRA (Dettmers et al., 2023). This work falls under PTQ as it does not require further training of a model.

**Numeric Representation** While integer quantization is the most commonly used numeric representation, other numeric representations, that have been explained above, are also used for inference and/or training: fp8 (Wang et al., 2018), fp6 (Gernigon et al., 2023), fp4 (Sun et al., 2020), nf4, and af4 (Yoshida, 2023).

**Lookup Table (LUT) Representation** While most research quantize to pre-defined numeric formats, other approaches use a dynamic format that is specified for each tensor or subset of elements of a tensor using a look-up-table (LUT) (a.k.a. codebook). In scalar quantization techniques, e.g., DeepCompression for CNNs (Han et al., 2016), GOBO for BERT (Zadeh et al., 2020), SqueezeLLM for LLMs (Kim et al., 2023), LUTs map scalar quantized values to scalar high precision values. In vector quantization techniques (Stock et al. for CNNs (Stock et al., 2020), AQLM for LLMs (Egiazarian et al., 2024)), LUTs map vectors of quantized values to vectors of high precision values.

**Preserving Outlier/Sensitive Values** LLM.int8() (Dettmers et al., 2022a) found that keeping $< 0.1\%$ of outlier activations and their corresponding weights in high precision minimizes drop in accuracy. SqueezeLLM (Kim et al., 2023) found that keeping 0.40% outlier weights and an additional 0.05% sensitive weights, determined by a Hessian metric, minimizes accuracy drops. In this work, we quantize all values and keep no outlier/sensitive values in higher precision.

**Pre-processing Weights and/or Activations** While many quantization algorithms simply round each high precision value to a value in the quantized set of possible values (Round to Nearest (RTN), stochastic rounding (Xia et al., 2021), or adaptive rounding (Nagel et al., 2020)), other algorithms perform some offline or online processing of weights and/or activations. Instead of keeping outlier activations or sensitive weights, AWQ (Lin et al., 2024) and SmoothQuant (Xiao et al., 2023) mitigate their effects by dividing outlier channels by a scaling factor and compensating by multiplying weights with the same factor. Other quantization approaches mitigate outliers by applying matrix transformations on weights and activations, e.g., QuIP (Chee et al., 2023), QuaRot (Ashkboos et al., 2024a) and SpinQuant (Liu et al., 2024). Another line of research follows an iterative procedure of quantizing weights in subsets, modifying unquantized elements to mitigate the errors introduced after quantizing each subset, e.g., GPTQ (Frantar et al., 2023).

A common trend is to use a combination of techniques. QuIP cascades incoherence processing with adaptive rounding, QTIP (Tseng et al., 2024) uses Hadamard transforms to remove outliers, vector quantization for numeric representation and other techniques, while SqueezeLLM preserves a portion of outlier/sensitive values in high precision and applies scalar quantization. In this work, we opt for a one-shot quantization algorithm that does not require any online or offline pre-processing or transformations on weights and/or activations, and focus on the aspect of learning quantization from data with efficient inference in hardware, achieving SOTA accuracies compared to other numeric format approaches and is competitive with orthogonal approaches that pre-process weights and activations. We leave it to future work to combine any4 with such orthogonal techniques.

## 4. Proposed Solution

### 4.1. any4 Algorithm

In any4 quantization, we first apply group-wise scaling, then try to find the optimal numeric representation for each row of a weight matrix. Naively applying K-means clustering on scaled weights will lead to a sub-optimal quantization scheme. This is because K-means clustering will minimize the reconstruction error of the weight matrix rather than the output of multiplying weights with sample inputs, and even for weight reconstruction, K-means clustering will minimize the reconstruction error of the scaled weight matrix rather than the original weight matrix.

We denote a weight matrix with dimensions of $N \times K$ as $\boldsymbol{w}$, an input vector with dimensions $M \times K$, where $M = 1$ without loss of generality, as $\boldsymbol{x}$, and the output vector with dimensions $M \times N$ as $\boldsymbol{y}$. Matrix multiplication in high

precision can be expressed as:

$$y = wx \quad (5)$$

and matrix multiplication with quantized weights as:

$$\hat{y} = \text{dequant}(w_Q)x \quad (6)$$

For the $i$th element of output $y$, this is equivalent to:

$$y_i = \sum_{\forall j} w_{i,j} x_j \quad (7)$$

$$\hat{y}_i = \sum_{\forall j} \text{dequant}(w_{Q_{i,j}}) x_j \quad (8)$$

Our goal is to find the set of $2^n$ quantized values for row $i$:

$$Q_i = \{w_{Q_i^0}, w_{Q_i^1}, \ldots, w_{Q_i^{2^n-1}}\} \quad (9)$$

for $n$-bit quantization (any$n$) that will minimize the expected mean square error in output activations for possible input activations:

$$\min_{Q_i} \mathbb{E} \|\hat{y} - y\| \quad (10)$$

We choose a greedy approach to minimize the mean of Frobenius norm of the error of the output activation vector by minimizing the absolute error of each of its elements:

$$\min_{Q_i} \mathbb{E} |\hat{y}_i - y_i| = \min_{Q_i} \mathbb{E} \left| \sum_{\forall j} w_{i,j} x_j - \sum_{\forall j} \text{dequant}(w_{Q_{i,j}}) x_j \right|$$

$$= \min_{Q_i} \mathbb{E} \left| \sum_{\forall j} (w_{i,j} - \text{dequant}(w_{Q_{i,j}})) x_j \right| \quad (11)$$

This way, we can focus on dealing with finding the optimal quantization configuration for each row i of the weight matrix. (Note that GPTQ opts to minimize output activations error in a different way such that all rows of the weight matrix are co-optimized together). Expanding the right hand side of the equation:

$$\min_{Q_i} \mathbb{E} |\hat{y}_i - y_i| = \min_{Q_i} \mathbb{E} \left| \sum_{\forall j} (w_{i,j} - (\alpha_{i,j} w_{Q_{i,j}} + \beta_{i,j})) x_j \right| \quad (12)$$

The high precision weights are mathematically equivalent to applying scaling factors on scaled weights (i.e., re-arrange Eqn. 1 to expand $w_{i,j}$ into $w_{i,j} = \alpha_{i,j} w_{S_{i,j}} + \beta_{i,j}$):

$$\min_{Q_i} \mathbb{E} |\hat{y}_i - y_i|$$

$$= \min_{Q_i} \mathbb{E} \left| \sum_{\forall j} (\alpha_{i,j} w_{S_{i,j}} + \beta_{i,j} - (\alpha_{i,j} w_{Q_{i,j}} + \beta_{i,j})) x_j \right|$$

$$= \min_{Q_i} \mathbb{E} \left| \sum_{\forall j} (\alpha_{i,j} (w_{S_{i,j}} - w_{Q_{i,j}}) x_j \right| \quad (13)$$

The offset factors, $\beta_{i,j}$, cancel each other out. Hence, we have:

$$\min_{Q_i} \mathbb{E} |\hat{y}_i - y_i| = \min_{Q_i} \mathbb{E} \left| \sum_{\forall j} (\alpha_{i,j} w_{S_{i,j}} x_j - \alpha_{i,j} w_{Q_{i,j}} x_j) \right| \quad (14)$$

We now proceed to solve this by a K-Means-style alternating optimization procedure:

0. **Initialize:** for $i$th row of a weight matrix, randomly initialize a set $Q_i$ to a random set of $2^n$ values:

$$Q_i = \{w_{Q_i^0}, w_{Q_i^1}, \ldots, w_{Q_i^{2^n-1}}\} \quad (15)$$

1. **E-Step:** Given $Q_i$ and the row of scaled weights:

$$\{w_{S_{i,j}}\}_{\forall j} = \{w_{S_{i,0}}, w_{S_{i,1}}, \ldots, w_{S_{i,M-1}}\} \quad (16)$$

we would like to deduce the best $w_{Q_{i,j}}$ for each corresponding $w_{S_{i,j}}$ that will minimize the expression defined in Eq. 14. Since in this step, the possible values in $Q_i$ are fixed and we are merely selecting from a set of discrete values, we apply a local minimization step and re-write Eq. 14 to:

$$w_{Q_{i,j}} = \min_{w_{Q_{i,j}} \in Q_i} (\alpha_{i,j} w_{S_{i,j}} x_j - \alpha_{i,j} w_{Q_{i,j}} x_j)^2$$

$$= \alpha_{i,j} x_j \min_{w_{Q_{i,j}} \in Q_i} (w_{S_{i,j}} - w_{Q_{i,j}})^2 \quad (17)$$

Again since $\alpha_{i,j} x_j$ are fixed in this step and are independent of $w_{Q_{i,j}}$, we can drop that term:

$$w_{Q_{i,j}} = \min_{w_{Q_{i,j}} \in Q_i} (w_{S_{i,j}} - w_{Q_{i,j}})^2 \quad (18)$$

2. **M-Step:** After applying the E-Step above, each $w_{Q_{i,j}}$ will be set to one of the $2^n$ values in the set $Q_i$. We refer to each set of indices $i, j$ that are associated with a specific quantized value $Q_i^q$ as a cluster. We can re-write Eq. 14 to create a separate sum term for elements in each cluster:

$$\min_{Q_i} \mathbb{E} |\hat{y}_i - y_i|$$

$$= \min_{Q_i} \mathbb{E} \left| \sum_{\forall j} \sum_{\forall q \in Q_i^q} (\alpha_{i,j} w_{S_{i,j}} x_j - \alpha_{i,j} w_{Q_i^q} x_j) \right| \quad (19)$$

To minimize the term, we can aim to set the difference for elements for each cluster to 0:

$$\mathbb{E} \left| \sum_{\forall q \in Q_i^q} (\alpha_{i,j} w_{S_{i,j}} x_j - \alpha_{i,j} w_{Q_i^q} x_j) \right| = 0 \quad (20)$$

The expression inside the expectation operation is a scalar value. Moreover, except for input activations $x$, all the other variables are deterministic and known offline. Hence, the expectation operator is only needed to be applied on input activations:

$$\sum_{\forall q \in Q_i^q} \left( \alpha_{i,j} w_{S_{i,j}} \mathbb{E} \left| x_j \right| - \alpha_{i,j} w_{Q_i^q} \mathbb{E} \left| x_j \right| \right) = 0 \quad (21)$$

Re-writing:

$$\sum_{\forall q \in Q_i^q} \alpha_{i,j} w_{S_{i,j}} \mathbb{E} \left| x_j \right| = \sum_{\forall q \in Q_i^q} \alpha_{i,j} w_{Q_i^q} \mathbb{E} \left| x_j \right| $$
$$= w_{Q_i^q} \sum_{\forall q \in Q_i^q} \alpha_{i,j} \mathbb{E} \left| x_j \right| \quad (22)$$

Re-arranging:

$$w_{Q_i^q} = \frac{\sum_{\forall q \in Q_i^q} \alpha_{i,j} w_{S_{i,j}} \mathbb{E} \left| x_j \right|}{\sum_{\forall q \in Q_i^q} \alpha_{i,j} \mathbb{E} \left| x_j \right|} \quad (23)$$

Eqn. 23 states that the optimal value to represent a group of scaled weights within a cluster is their average weighted by the product of the scaling factor of a weight element and mean of the norm of activations applied to that element.

We alternate between the E-Step and M-Step till the values of $Q_i$ converge.

The equation of E-Step is equivalent to the cluster assignment step of K-means clustering, while the equation of M-Step is equivalent to the centroid update step of weighted K-means. Hence, our mathematical formulations guides us to creating the LUT of each row of a scaled weight matrix by the algorithm depicted in Alg. 1. We also summarize our algorithm in Fig. 2. We speedup the process by parallelizing the loop over each linear weight's rows, enabling us to quantize Llama3 8B in 10 minutes.

While most quantization papers use a dataset like C4 to obtain a set of calibration activations, we hand curate a single calibration sample, as shown in Listing. 1, that covers diverse set of topics, and then obtain the mean of absolute of activations along the channel axis to represent $\mathbb{E} \left| x \right|$.

- Fiction: "Once upon a time, a girl named Alice was living alone on an island. One day, she met a wizard ..."
- News: "The United Nations held its General Assembly meeting this year amid multiple world crises and wars. In his speech, the General Secretary called for ..."
- Code: ˜public static void main(String[] args) \n System.out.println("Hello world!");\n ˜
- Math: (5.2 + 2.7) / 0.6 - 1.9 * 2.2 =
- Facts: "The capital of Egypt is Cairo. It is the largest city in the region and is home to..."

Listing 1: Calibration sample used to generate LUTs.

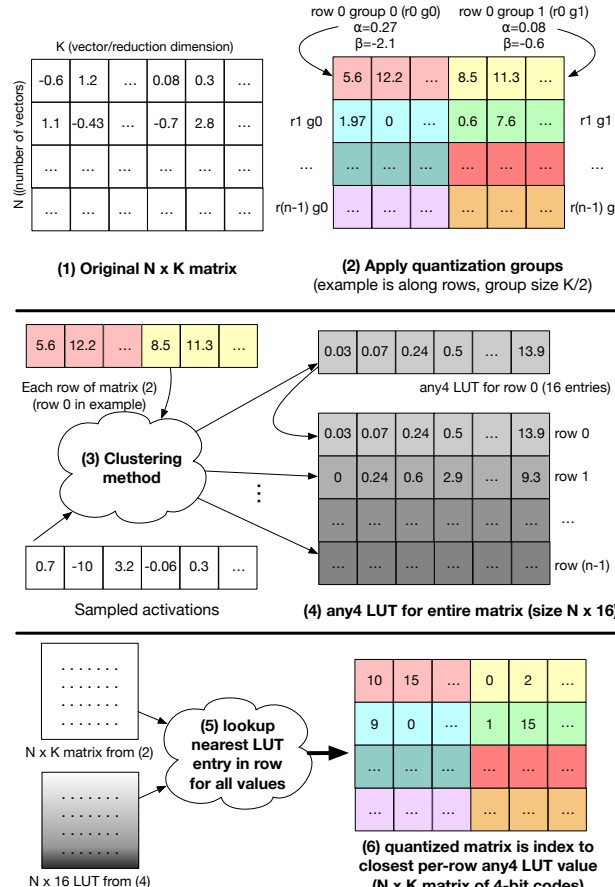

Figure 2: any4 quantization process

### 4.2. tinygemm Library

As part of this paper, we present tinygemm, a GEMM library optimized for low-latency LLM inference at small batch sizes (1 to 16) for Nvidia GPU Ampere generation and later architectures. For a matrix multiplication $\boldsymbol{y} = \boldsymbol{x}\boldsymbol{w}^T$ where $\boldsymbol{x}$ is of size $M \times K$ and $\boldsymbol{w}$ is of size $N \times K$ ($M$ and $N$ being the *outer* dimensions and $K$ being the *reduction* dimension), in linear layers, the product of batch size

| | **Perplexity ↓** | | | | **Tasks ↑** | | | | | |
|---|---|---|---|---|---|---|---|---|---|---|
| | **WikiText-2** | **C4** | **PTB** | **CodeParrot** | **HumanEval Pass@1** | **MBPP Pass@1** | **MMLU** | **HellaSwag** | **GSM8K** | **BBH** |
| **Llama3.2 1B** | | | | | | | | | | |
| FP16 | 9.76 | 12.77 | 16.56 | 3.49 | 16.46% | 21.4% | 36.1% | 47.7% | 6.60% | 31.1% |
| INT4 | 11.89 | 15.74 | 20.32 | 4.08 | 9.76% | 11.4% | 30.1% | 44.7% | 3.18% | 26.2% |
| FP4 | 13.01 | 17.11 | 21.89 | 4.28 | 8.54% | 5.8% | 29.3% | 43.6% | 2.27% | 23.3% |
| NF4 | 10.99 | 14.63 | 18.78 | 3.82 | **13.4%** | 13.8% | **33.3%** | 45.8% | 2.65% | 26.8% |
| ANY4 | **10.63** | **13.95** | **17.94** | **3.71** | 11.0% | **18.6%** | 32.9% | **46.7%** | **3.71%** | **29.0%** |
| **Llama3 8B** | | | | | | | | | | |
| FP16 | 6.14 | 8.93 | 10.59 | 2.54 | 29.3% | 41.4% | 62.0% | 60.1% | 50.7% | 62.8% |
| INT4 | 6.87 | 9.89 | 11.37 | 2.83 | 23.2% | 35.4% | 59.6% | 58.6% | 40.6% | 58.5% |
| FP4 | 7.10 | 10.22 | 11.81 | 2.89 | 22.0% | 36.8% | 57.1% | 58.5% | 35.0% | 53.2% |
| NF4 | 6.63 | 9.52 | 11.14 | 2.72 | **23.2%** | **39.2%** | 60.7% | 59.1% | 41.1% | 59.0% |
| ANY4 | **6.51** | **9.40** | **11.07** | **2.68** | 21.3% | **39.2%** | **61.0%** | **59.5%** | **41.7%** | **59.2%** |
| **Llama3 70B** | | | | | | | | | | |
| FP16 | 2.86 | 6.77 | 8.16 | 1.91 | 17.7% | 60.8% | 75.4% | 66.3% | 80.6% | 82.4% |
| INT4 | 3.63 | 7.97 | 8.86 | 2.21 | 18.3% | 45.0% | 73.0% | **66.2%** | 73.9% | 78.4% |
| FP4 | 3.94 | 7.76 | 8.99 | 2.17 | **22.0%** | 50.8% | 71.9% | 65.6% | 75.3% | 77.9% |
| NF4 | 3.43 | 7.67 | 8.84 | 2.15 | 18.9% | 39.6% | 73.7% | 66.1% | 75.9% | 79.3% |
| ANY4 | **3.20** | **7.01** | **8.33** | **1.99** | 17.1% | **57.4%** | **75.1%** | 66.1% | **78.5%** | **81.8%** |

Table 1: Quantizing Llama3 models with various numeric formats. Results for Llama2 and Mistral/Mixtral are in the Appendix.

and sequence length corresponds to matrix dimension $M$. At $M \leq 8$, activation $x$ is itself much smaller than tensor core tile sizes ($m = 16, n = 8, k = 16$) for 16-bit float Ampere+ mma "tensor core" fixed-function matrix multiplication instructions. In this case, each $8 \times 16$ tile of $w$ (weights) is only used once (no data reuse). Thus, multistage asynchronous pipelining and data reuse concerns in typical high-performance GPU GEMM kernels are reduced, as the problem is largely memory latency (or bandwidth) limited. Tensor cores still outperform manual (scalar) matrix multiplication at $M = 1$ (GEMV / matrix-vector multiplication) per our analysis. An early version of tinygemm, largely focused on int4 grouped quantization for small batch sizes, has been part of core PyTorch since late 2023, subsequently utilized by gpt-fast (PyTorch, 2023), torchao (PyTorch, 2024), and Hugging Face Transformers (Wolf et al., 2020).

Many inference works (especially in open source) concentrate on $M = 1$ performance, where latency is a concern. Even in this case, where we would be using only $\frac{1}{8}$ or $\frac{1}{16}$ of tensor core throughput, we improve latency by laying out matrices in main (global) memory in the exact format that mma expects per tile rather than standard row-major / column-major format. Typical tensor core GEMM kernels use shared memory (a small, high-speed user-controllable scratchpad memory) to transpose tiles of matrices into the desired format before multiplication can proceed. We avoid this by performing the transposition in advance, allowing matrix data to pass directly from global memory to registers.

As there is little to no weight reuse opportunity for small batch sizes, and loads into registers can be asynchronous as they generally do not stall execution until the point of first use, tinygemm does not use shared memory in many cases. This strategy improves performance at small batch sizes, but is not applicable for larger sizes. To improve efficiency, when $M \leq 8$, we maintain weights on the left to use the $16 \times 16$ tile, computing $y = (wx^T)^T$ flipping the order of matrices presented to mma with transpositions performed on the fly, and if $M > 8$, we maintain weights on the right for the $8 \times 16$ tile ($y = xw^T$).

To implement int4, nf4, or any4 GEMM, we dequantize weights on the fly before mma multiplication. Speed is improved by always ensuring that we can load matrix data using vectorized 16 byte loads in coalesced and contiguous fashion across the warp from global memory. In cases where a single thread's quantized tile data is less than 16 bytes (a m16n8k16 "B" tensor core layout with quantized 4-bit values only needs 2 bytes loaded prior to dequantization per CUDA thread per mma), multiple tiles along the reduction dimension ("$k$-tiles" in tinygemm terminology) can be packed together to ensure that wide data loads can be used in all cases.

Instead of typical int4-to-float dequantization (converting an integer in [-8, 7] to floating point via native instructions or bit manipulation), we can use a 16-entry LUT per row containing arbitrary floating point values. In tinygemm, this LUT is held in a single register with lookup provided using

| | Quantization Algorithm | Numeric Format | WikiText-2 Perplexity ↓ | | Numeric Format | WikiText-2 Perplexity ↓ | | Numeric Format | WikiText-2 Perplexity ↓ |
|---|---|---|---|---|---|---|---|---|---|
| **Llama3 8B** | | | | | | | | | |
| | FP16 | | 6.1 | | | | | | |
| | RTN | INT4 | 6.9 | | INT3 | 17.1 | | INT2 | 1.9E3 |
| | GPTQ | INT4 | **6.5** | | INT3 | 8.2 | | INT2 | 2.1E2 |
| **4-bits** | AWQ | INT4 | 6.6 | **3-bits** | INT3 | 8.2 | **2-bits** | INT2 | 1.7E6 |
| | QuIP | INT4 | **6.5** | | INT3 | **7.5** | | INT2 | **85.1** |
| | RTN | ANY4 | **6.5** | | ANY3 | 8.0 | | ANY2 | 1.0E3 |
| **Llama3 70B** | | | | | | | | | |
| | FP16 | | 2.9 | | | | | | |
| | RTN | INT4 | 3.6 | | INT3 | 11.8 | | INT2 | 4.6E5 |
| | GPTQ | INT4 | 3.3 | | INT3 | 5.2 | | INT2 | **11.9** |
| **4-bits** | AWQ | INT4 | 3.3 | **3-bits** | INT3 | 4.8 | **2-bits** | INT2 | 1.7E6 |
| | QuIP | INT4 | 3.4 | | INT3 | 4.7 | | INT2 | 13.0 |
| | RTN | ANY4 | **3.2** | | ANY3 | **4.6** | | ANY2 | 253.8 |

Table 2: Quantizing Llama3 models with various quantization algorithms for different bit widths.

GPU warp shuffle functionality, with the 4-bit quantization codes used as LUT indices. An alternative strategy would be to use a shared memory LUT containing all possible $16 \times 16 = 256$ pairs of any4 reproduction values so that two packed any4 values (in a byte) can be dequantized per lookup. While this amount of shared memory usage will likely not affect performance (via occupancy) that much, it does suffer shared memory bank conflict penalties in many circumstances.

## 5. Results

We quantize weights of all linear modules of all transformer layers: key, query, value, and output projections, up, down projections and gate for feed-forward networks (FFN). Following most quantization papers, we keep weights of embedding and final classification layers high-precision.

We evaluate both perplexity and downstream tasks. For perplexity, we ported the implementation of GPTQ for WikiText-2 (Merity et al., 2017), C4 (Raffel et al., 2019), and Penn Treebank (Marcus et al., 1993) that is used by codebases of other quantization papers. To add coding domain, we added perplexity on CodeParrot (CodeParrot).

For downstream tasks, we used Eleuther Harness (Gao et al., 2024) for natural language tasks, and BigCode Harness (Ben Allal et al., 2022) for coding tasks. Accuracies on downstream tasks tend to be noisy (Wang et al., 2024), while perplexity is a less noisy indicator of a model's performance.

**Comparison with Other Numeric Representations** We first compare accuracy of any4 with other numeric formats: int4, fp4, nf4. We use group-wise scaling with group size 128, and asymmetric scaling for all models, except for Llama3 70B where we found symmetric scaling leads to

better results.

We ran on different model families (Llama (Touvron et al., 2023a) and Mistral (Jiang et al., 2023)), different generations (Llama2 (Touvron et al., 2023b) and Llama3 (Grattafiori et al., 2024)), and different sizes (from 1B all the way to 70B). We provide results of Llama3 in Table 1, Llama2 in Table A1, and Mistral in Table A2. Our results show any4 has the best accuracies across all models.

**Speed Comparisons** We benchmark matrix multiplication of vector activation and square weight tensors from 1K to 16K on A100 80GB GPU using PyTorch 2.3.0 and provide the speedups of our tinygemm library in Fig. 3. int4, nf4, and any4 were implemented using our tinygemm library. int4 kernels have the highest speedup, reaching close to $3\times$. nf4 and any4 speedups reach up to $2\times$; lower than int4 because of the overhead of looking up the LUTs. Nevertheless, any4 has almost the same speedup as nf4, despite the latter requiring a single LUT for a whole tensor and the former requiring a separate LUT for each row in the weight matrix.

**Comparison with Orthogonal Quantization Techniques** As explained in the Related Works section, our work proposes a new numeric representation applying RTN (round-to-nearest). Despite our work being orthogonal to others that transforms weights and/or activations to make them more rounding or quantization friendly, we compare any4 to GPTQ, AWQ, and QuIP that use int4 in Table 2. Results of AWQ, GPTQ, and QuIP are obtained from (Huang et al., 2024). In 4-bit the results show that any4 has either the best or competitive performance. For future work, we can evaluate these orthogonal techniques together, replacing the int4 representation with any4.

**3-bit and 2-bit Quantization** Although our main goal was 4-bit representation, we ran experiments to see how any3 and

| | Llama3.2 1B | | | | | | |
|---|---|---|---|---|---|---|---|
| | **Calibration Data** | **Number of Samples** | **Sequence Length per Sample** | **Perplexity ↓** | | | |
| | | | | **WikiText-2** | **C4** | **PTB** | **CodeParrot** |
| FP16 | | | | 9.76 | 12.77 | 16.56 | 3.49 |
| ANY4 | WikiText-2 | 128 | 2048 | 10.70 | 14.08 | 18.02 | 3.74 |
| ANY4 | Pile | 128 | 2048 | 10.70 | 13.99 | 18.26 | 3.74 |
| ANY4 | C4 | 128 | 4096 | 10.74 | 14.14 | 18.10 | 3.75 |
| ANY4 | C4 | 128 | 2048 | 10.67 | 14.05 | 17.97 | 3.74 |
| ANY4 | C4 | 128 | 512 | **10.62** | 13.96 | 18.03 | 3.72 |
| ANY4 | Handwritten Prompt | 1 | - | 10.63 | **13.95** | **17.94** | **3.71** |

Table 3: any4 quantization with different calibration data.

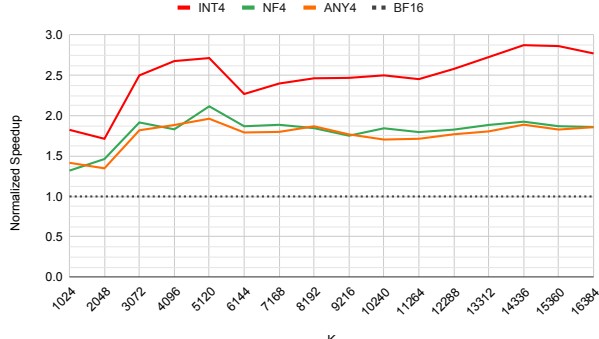

Figure 3: Speedup of our tinygemm CUDA kernels on 80GB A100 on matrix multiplication of $1 \times K$ input by $K \times K$ weight, w.r.t PyTorch's bfloat16 implementation.

any2 perform compared to the prior orthogonal quantization techniques (Table 2). For 3-bit, any3 is either the best or competitive with other approaches. For 2-bit, QuIP is the best, while any2 is better than AWQ and competitive with GPTQ.

### 5.1. Ablation Studies

**Calibration Data**

In Table 3 we ablate with different calibration datasets to calculate sample weighting in Eqn. 23 of our any4 algorithm. The results show that our proposed handwritten sample performs better than commonly used datasets in literature, despite being significantly smaller in number of tokens. Note that the handwritten sample or prompt has a fixed number of words that translates to different number of tokens depending on the tokenizer that changes with different models. Our prompt has 88 words only, which will in worst case translate to a few hundred tokens. These results may indicate that a single data sample with diverse topics could be enough or better to calibrate than using many long sample sequences. Our evaluation sequence length is 2048 (following (Lin et al., 2024; Frantar et al., 2023)), calibration is on training split of each dataset, and evaluation is on the validation or test split.

**Group Size** In Table 4 we ablate quantization group size from 64 to 1024. any4 always has the lowest perplexity across other 4-bit representations across all group sizes. It is noteworthy that fp4 and nf4 perplexity degenerates for large group sizes at 1024, while any4 only increases marginally.

| | Llama3.2 1B | | | | |
|---|---|---|---|---|---|
| | **Group Size** | | | | |
| | **64** | **128** | **256** | **512** | **1024** |
| FP16 | | | 12.77 | | |
| FP4 | 16.19 | 17.11 | 18.12 | 20.43 | 2.3E6 |
| NF4 | 14.27 | 14.63 | 14.98 | 15.38 | 7.8E5 |
| ANY4 | **13.75** | **13.95** | **14.09** | **14.24** | **14.34** |

Table 4: C4 perplexity after quantizing with different group sizes.

## 6. Conclusion & Future Work

We have presented any4, an algorithm to find an optimal low-bit numeric representation for each row in a weight matrix, as well as tinygemm, a matrix multiplication library for low-latency, low-bit inference. We have shown that accuracy of any4 is superior to other 4-bit numeric formats with low memory overhead, and competitive with various orthogonal quantization techniques that involve further pre-processing. We would like to explore combining with these orthogonal techniques in the future.

## Acknowledgements

We would like to thank Newsha Ardalani for help in running experiments; Daniel Haziza, Francisco Massa, Luca Wehrstedt, Bram Wasti, Steven Li, and Lin Xiao for discussions.

## Impact Statement

This paper presents a work that quantizes pretrained models. The input to the algorithm is a model's pretrained weights,

architecture, and a calibration dataset (which in our case was a single hand-written prompt). We have not evaluated if the quantization algorithm increases or decreases any societal impact of the underlying model. One factor that may introduce bias into the model is the calibration dataset. We leave it for future work to analyze the effect of different calibration datasets (or prompts in our case) on bias and truthfulness.

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

# Appendix

# A. Solution Details

We provide here more details about our proposed any4 algorithm.

### A.1. Algorithm

We summarize our any4 quantization algorithm in Alg. 1.

---

**Algorithm 1** any4 quantization algorithm.

---

```
module2input = calibrate(model, sample_data)

for module in model:
  w = module.weight()
  wQ = torch.zeros_like(w)
  alpha = []
  beta = []
  for i in range(w.shape[0]):
    wSi, alphai, betai = scale(w[i,:])
    xi = module2input[module][i]
    wQ[i, :] = kmeans(
        samples=wSi,
        sample_weight=alphai*abs(xi.mean())
    )
    alpha.append(alphai)
    beta.append(betai)
  module.weight.data = wQ
  module.alpha = alpha
  module.beta = beta
```

---

# B. Further Results

### B.1. Comparison with Other Numeric Formats

We compare our any4 numeric format with other numeric formats for the Llama2 family of models in Table A1 and for Mistral-7B and Mixtral-7B in Table A2.

| | | Perplexity ↓ | | | | Tasks ↑ | | |
|---|---|---|---|---|---|---|---|---|
| | **WikiText-2** | **C4** | **PTB** | **CodeParrot** | **MMLU** | **HellaSwag** | **GSM8K** | **BigBench** |
| **Mistral-7B Instruct v0.2** | | | | | | | | |
| FP16 | 5.95 | 8.82 | 21.77 | 2.63 | 58.7% | 66.1% | 41.7% | **51.7%** |
| INT4 | 6.14 | 9.03 | 22.02 | 2.70 | 57.1% | 65.1% | 39.7% | 50.4% |
| FP4 | 6.19 | 9.10 | 21.62 | 2.70 | 56.6% | 64.7% | 38.2% | 47.7% |
| NF4 | 6.06 | 8.93 | 24.72 | 2.66 | 58.0% | **65.5%** | 38.5% | **51.8%** |
| ANY4 | **6.00** | **8.85** | **23.24** | **2.64** | **58.6%** | 65.4% | **41.1%** | 51.7% |
| **Mixtral-8x7B Instruct v0.1** | | | | | | | | |
| FP16 | 4.14 | 7.18 | 16.47 | 2.20 | 68.2% | 67.6% | 64.8% | 68.1% |
| INT4 | 4.45 | 7.45 | 16.84 | 2.26 | 66.5% | 66.3% | 57.8% | 61.8% |
| FP4 | 4.46 | 7.48 | 18.42 | 2.27 | 66.8% | 66.5% | 59.4% | 62.8% |
| NF4 | 4.30 | 7.32 | **15.00** | 2.24 | 67.6% | **67.2%** | 61.0% | **66.5%** |
| ANY4 | **4.27** | **7.27** | 16.14 | **2.22** | **67.7%** | 67.1% | **62.8%** | 65.8% |

Table A2: Quantizing Mistral and Mixtral with various numeric formats.

| | Perplexity ↓ | | | | Tasks ↑ | | | | | |
|---|---|---|---|---|---|---|---|---|---|---|
| | **WikiText-2** | **C4** | **PTB** | **CodeParrot** | **HumanEval Pass@1** | **MBPP Pass@1** | **MMLU** | **HellaSwag** | **GSM8K** | **BBH** |
| **Llama2 7B** | | | | | | | | | | |
| FP16 | 5.47 | 6.97 | 20.83 | 2.54 | 17.1% | 20.0% | 41.3% | 57.2% | 13.6% | 39.8% |
| INT4 | 5.74 | 7.30 | 24.00 | 2.63 | 10.4% | 18.2% | 38.1% | 56.4% | 10.6% | 36.5% |
| FP4 | 5.83 | 7.37 | 22.57 | 2.65 | 11.0% | 16.8% | 36.5% | 56.6% | 11.2% | 35.5% |
| NF4 | 5.66 | 7.19 | 22.82 | 2.60 | 11.6% | **19.2%** | 37.4% | **56.8%** | 12.0% | 36.8% |
| ANY4 | **5.59** | **7.10** | **21.23** | **2.57** | **14.0%** | 18.4% | **40.3%** | 56.7% | **12.7%** | **36.9%** |
| **Llama2 13B** | | | | | | | | | | |
| FP16 | 4.88 | 6.47 | 28.93 | 2.40 | 19.5% | 18.4% | 50.5% | 60.0% | 23.2% | 47.4% |
| INT4 | 5.05 | 6.65 | 30.79 | 2.45 | 15.2% | 16.4% | 48.8% | 59.3% | 20.8% | 44.2% |
| FP4 | 5.07 | 6.67 | 30.96 | 2.46 | 15.2% | 16.2% | 49.5% | 59.3% | 19.3% | 43.0% |
| NF4 | 4.99 | 6.58 | 31.17 | 2.43 | **15.9%** | 16.0% | **49.9%** | **59.9%** | **22.1%** | **44.6%** |
| ANY4 | **4.97** | **6.55** | **28.83** | **2.42** | 15.2% | **18.0%** | 49.3% | 59.5% | 21.6% | **44.6%** |
| **Llama2 70B** | | | | | | | | | | |
| FP16 | 3.32 | 5.52 | 14.44 | 2.11 | 31.7% | 37.4% | 65.2% | 64.8% | 53.3% | 67.1% |
| INT4 | 3.46 | 5.61 | 14.71 | 2.14 | 26.8% | **37.8%** | 64.4% | **64.7%** | 51.4% | 65.0% |
| FP4 | 3.53 | 5.67 | 14.34 | 2.16 | 28.0% | 30.6% | 64.1% | 64.0% | **51.6%** | 65.0% |
| NF4 | 3.44 | 5.61 | 14.65 | 2.14 | **29.9%** | 37.2% | 64.5% | 63.9% | 50.6% | 65.4% |
| ANY4 | **3.40** | **5.58** | 14.64 | **2.13** | 26.8% | 35.8% | **64.8%** | 64.5% | **51.6%** | **66.6%** |

Table A1: Quantizing Llama2 models with various numeric formats.

## C. Further Ablation Studies

### C.1. Minimization Terms

In Table A3 we ablate on using different terms to minimize when learning (using K-means clustering) the LUT of each row in the weight matrix. First row shows the results of optimizing weights directly. The other 2 rows show the results of using the 2 additional terms of Equation 14 in our paper, i.e., multiplying with activations and scales. These results confirm that our derivation that lead to all the terms of Equation 14 is essential for optimal accuracy.

| **Llama3.2 1B** | | | | | |
|---|---|---|---|---|---|
| | **Term to Minimize** | **WikiText-2** | **C4** | **PTB** | **CodeParrot** |
| Weights Only | $(w_{S_{i,j}} - w_{Q_{i,j}})$ | 6.680 | 9.619 | 11.186 | 2.751 |
| Weights × Activations | $(w_{S_{i,j}} x_j - w_{Q_{i,j}} x_j)$ | 6.496 | 9.375 | 11.055 | **2.675** |
| Weights × Activations × Group Scales [Ours] | $(\alpha_{i,j} w_{S_{i,j}} x_j - \alpha_{i,j} w_{Q_{i,j}} x_j)$ | **6.487** | **9.366** | **11.034** | 2.680 |

Table A3: Perplexity after quantizing Llama3.2 1B with LUTs created by minimizing different terms.

### C.2. K-Means Initialization

We use scikit (Pedregosa et al., 2011) to implement K-means clustering, that is core to any4's quantization algorithm. By default, scikit initializes cluster centroids using k-means++ algorithm (Arthur & Vassilvitskii, 2007), but it also supports random initialization, as well as initializing with a vector of pre-defined values. In Table A4 we ablate K-means initialization on Llama 3.2 1B by evaluating k-means++ and random initialization, as well as seeding with uniform int4 values (i.e., integer values -7 to 8), and nf4 values (ranging from -1 to +1). We see that k-means++ performs clearly the best, while uniform int4 initialization performs the worst.

| Llama3.2 1B | | | | |
|---|---|---|---|---|
| | **K-Means Initialization** | **Perplexity ↓** | | |
| | | **WikiText-2** | **C4** | **PTB** |
| FP16 | | 9.76 | 12.77 | 16.56 |
| ANY4 | k-means++ | **10.63** | **13.95** | **17.94** |
| ANY4 | random | 10.66 | 13.97 | 18.17 |
| ANY4 | int4 | 10.83 | 14.21 | 18.69 |
| ANY4 | nf4 | 10.65 | 13.96 | 18.21 |

Table A4: any4 quantization with K-means clustering initialzied with different algorithms and values.

