# OpenReview forum: "any4: Learned 4-bit Numeric Representation for LLMs"
_ICML.cc/2025/Conference — ICML 2025 poster_

### Official Review · Reviewer_U2Ld · 2025-03-07

**Overall Recommendation:** 2

**Summary:**

This paper introduces a 4-bit weight quantization method called any4, designed for Large Language Models (LLMs).  The authors claim that this method offers arbitrary numeric representations without the need for preprocessing weights or activations.  The paper compares any4 with other 4-bit numeric representation types like int4, fp4, and nf4, using various LLMs (Llama 2, Llama 3, Mistral, and Mixtral) of different sizes and families.  The key finding is that any4 outperforms other numeric formats in terms of accuracy.  The paper also demonstrates that any4 is competitive with techniques like AWQ and GPTQ, which require weight/activation preprocessing.  Additionally, the authors show the competitiveness of any3 and any2 at lower bit widths and highlight the ability to calibrate any4 using a single, curated diverse sample instead of hundreds of dataset samples.  The paper also introduces tinygemm, an open-source GPU matrix multiplication library optimized for LLMs.  tinygemm efficiently implements any4 using a lookup table strategy on GPUs.

**Claims And Evidence:**

Overall, the paper presents a compelling case for the any4 quantization method. The claims are generally well-supported by evidence from experiments and comparisons with existing techniques. However, there are a few areas where further clarification or investigation could strengthen the paper:

Strengths:

- Clear evidence of any4's superior accuracy compared to {int4, fp4, and nf4} in terms of perplexity and downstream task accuracy.
- Competitive performance with preprocessing techniques such as {AWQ, GPTQ, and QuIP}
- Effectiveness at lower bit widths (any3)
- Efficient calibration: The ability to calibrate any4 with a single, diverse sample is a significant advantage.
- Open-source implementation: The tinygemm library provides a practical implementation of any4, allowing for further research and adoption.

Areas for improvement:

- Calibration sample details: the paper shows using a single, curated diverse sample for calibration, but does not explain how it was constructed or why they thought this was sufficient. More information about this process would be beneficial.
- Further exploration of any4 with other techniques: While the paper demonstrates any4's competitiveness with existing techniques, it doesn't explore combining any4 with those techniques (e.g., using any4 as the numeric format for GPTQ or AWQ). This could be a promising avenue for future work.
- Any2: the paper claims any3 and any2 are the effective, but looking at the results, any2 doesn't really work...


Despite these minor points, the paper provides strong evidence to support its claims and makes a valuable contribution to the field of LLM quantization.

**Essential References Not Discussed:**

not that i can immediately think of

**Experimental Designs Or Analyses:**

Yes

Experimental Design:
- They used a wide range of models (Mistral, Llama 2, Llama 3), and sizes (from 1B to 70B)
- They compared with a good set of numeric formats (int4, fp4, nf4) and quantization techniques (AWQ, GPTQ, QuIP)/
- They ablate different group sizes

Analysis:The paper evaluates speedup vs perplexity and downstream tasks

Potential Issues:
- limited exploration of combining any4 with e.g. GPTQ or AWQ

Despite these minor points, the experimental designs and analyses in the paper are generally sound and provide convincing evidence to support the claims made about the any4 quantization method.

**Methods And Evaluation Criteria:**

Yes
The paper made sensible choices (row-wise grouping), focused on downstream accuracy, showed performance across a large set of model sizes, compared to other low precision quant techniques, and had an easy calibration setup. This makes sense for the problem or application at hand.

**Other Comments Or Suggestions:**

This is just a learned compression codebook (LUT) which optimizes for reconstructing the output (instead of minimizing mse of weights error).
I've personally talked about this with 2 separate orgs; in my mind its not novel, but I guess I cant think of anyone who has published it already...

**Other Strengths And Weaknesses:**

The work shows improved performance, but does have performance degradation. Its fast but this is on old hardware (A100s). When B200s come out with native fp4 support, this will not work anymore (unless you can write fp4 codebook with your learned codebook).

**Questions For Authors:**

none

**Relation To Broader Scientific Literature:**

Accelerating llm inference via compressed representations is an active area of research in the community.

**Theoretical Claims:**

Yes.
The theoretical claim in the paper:
```
Eqn. 23 states that the optimal value to represent a group of scaled weights within a cluster is their average weighted by the product of the scaling factor of a weight element and mean of the norm of activations applied to that element.
```
This just follows from their equations.

---

> ### Author Rebuttal · Authors · 2025-03-31
>
> We thank the reviewer for their constructive feedback, particularly recognizing that the paper **“presents a compelling case”**, and **“provides strong evidence to support its claims and makes a valuable contribution to the field of LLM quantization”**, as well as the important suggestions to improve the quality of the paper.
>
> We address the reviewer’s comments as follows:
> - **Calibration Sample Construction:** The calibration text sample (that we have provided in Section A.2 in the Appendix) was manually crafted with sentences spanning diverse domains: Fiction, News, Code, Math, Facts, and concatenating them together. Our intuition was that having a single sample with many diverse topics would be sufficient, rather than extracting many samples from a Common Crawl dataset and hoping that they will cover diverse domains.
> - **Combining any4 with AWQ or GPTQ:** We have attempted to combine any4 with AWQ during the rebuttal period and provided results in Table B (in response to Reviewer pDZ9) in this rebuttal. We show that indeed combining AWQ with ANY4 improves the results of AWQ.
> - **Native FP4 Support in New Nvidia B200 GPUs:** We acknowledge that new hardware is supporting FP4 as a native format for computation but we emphasize that our approach is particularly **effective for PTQ** and **can enable arbitrary number of bits**:
>   - In our experiments, we observed that FP4 leads to lower Post-Training Quantization (PTQ) accuracy compared to NF4 and our proposed ANY4 format. i.e., FP4 does not perform well for zero-shot quantizing without training.
>   - Nevertheless, FP4 works well for Quantized Aware Training (QAT). Native support for computing in FP4 enables training models faster with FP4. However, our paper explicitly focuses on post-training quantization (PTQ), a practical setting where model weights are quantized without retraining.
>   - Furthermore, our method also achieves strong results with ANY3, a 3-bit format for which there is currently no native hardware support—highlighting the broader applicability of our lookup-table-based quantization approach beyond current hardware capabilities. Moreover, it could be used for 6-bit quantization [H] which other papers are exploring.
> - **Clarification on ANY2:** We agree ANY2 is not competitive at this stage; we included it to illustrate the extensibility of our method across bit widths, even at the extreme low end.
>   - Moreover, QuIP which performs well on 2-bit quantization is orthogonal to our approach and hence may be integrated in future work to enhance 2-bit quantization performance.
> - **Novelty:** Our method is novel in that it is the first to apply scalar LUT-based quantization minimizing output error in LLMs without any finetuning, unlike prior work [F, G] that provided a formulation in reducing output error rather than weight error but required finetuning.
>
> **References**
>
> [E] “Ultra-Low Precision 4-bit Training of Deep Neural Networks”, Xiao Sun, Naigang Wang, Chia-Yu Chen, Jiamin Ni, Ankur Agrawal, Xiaodong Cui, Swagath Venkataramani, Kaoutar El Maghraoui, Vijayalakshmi (Viji) Srinivasan, Kailash Gopalakrishnan, https://papers.nips.cc/paper_files/paper/2020/hash/13b919438259814cd5be8cb45877d577-Abstract.html, NeurIPS 2020
>
> [F] “And the Bit Goes Down: Revisiting the Quantization of Neural Networks”, Pierre Stock, Armand Joulin, Rémi Gribonval, Benjamin Graham, Hervé Jégou, https://arxiv.org/abs/1907.05686, ICLR 2020
>
> [G] “Extreme Compression of Large Language Models via Additive Quantization”, Vage Egiazarian, Andrei Panferov, Denis Kuznedelev, Elias Frantar, Artem Babenko, Dan Alistarh, https://arxiv.org/abs/2401.06118, ICML 2024
>
> [H] "FP6-LLM: Efficiently Serving Large Language Models Through FP6-Centric Algorithm-System Co-Design", Haojun Xia, Zhen Zheng, Xiaoxia Wu, Shiyang Chen, Zhewei Yao, Stephen Youn, Arash Bakhtiari, Michael Wyatt, Donglin Zhuang, Zhongzhu Zhou, Olatunji Ruwase, Yuxiong He, Shuaiwen Leon Song, https://arxiv.org/abs/2401.14112

---

> > ### Comment · Reviewer_U2Ld · 2025-04-08
> >
> > Thank you for responding
> > I'm still not convinced that this is super novel. I'm boarderline weak accept / week reject, but am slightly leaning towards week reject.

---

### Official Review · Reviewer_aGNw · 2025-03-10

**Overall Recommendation:** 3

**Summary:**

The paper introduces "any4," a newly proposed learned 4-bit numeric representation aimed at optimizing the quantization of weights in large language models (LLMs). Any4 enhances accuracy compared to traditional 4-bit formats such as int4, fp4, and nf4, and does not require preprocessing of weights or activations. Furthermore, it shows competitive results against other techniques that perform such preprocessing, like Adaptive Weight Quantization (AWQ) and Generalized Post-Training Quantization (GPTQ). The authors also present tinygemm, a latency-optimized GPU matrix multiplication library designed for implementing any4 effectively. The results indicate that any4 achieves superior accuracy across various model sizes and types.

**Claims And Evidence:**

Overall, the claims about Any4 quantization are generally well supported by experimental results and methodological explanations. The performance improvements and efficiency gains are backed by perplexity results and design choices.

**Essential References Not Discussed:**

None

**Experimental Designs Or Analyses:**

It compares with other numeric formats (int4, fp4, nf4) and orthogonal quantization techniques (AWQ, GPTQ, QuIP). While the methodology is well-structured, the evaluation could be strengthened by ensuring statistical validation and consistent experimental conditions. Additionally, the ablation studies, currently in the Appendix, provide valuable insights and would be more impactful if moved into the main content for better visibility and discussion.

**Methods And Evaluation Criteria:**

The proposed methods and evaluation criteria in the paper for the any4 quantization technique are appropriate for addressing the challenges in optimizing large language models (LLMs).

1. Methodology: It employs group-wise scaling and K-means clustering for quantization, aiming to improve efficiency and accuracy in weight representation.

2. Evaluation Metrics: Using perplexity and assessing downstream task performance provide a reliable indication of the model's effectiveness in real-world applications.

3. Benchmark Datasets: The inclusion of diverse datasets (e.g., WikiText-2, C4) allows for comprehensive evaluation across different contexts, reinforcing the applicability of the proposed method.

Overall, the methods and evaluation criteria are well-tailored to the problem at hand, offering both theoretical advancement and practical usability in model deployment.

**Other Comments Or Suggestions:**

It has almost three pages of related work and background, which is too detailed and is not really necessary. This structure limits the space in the experiment section, and actually, ablation studies and other content can be moved to the main content. Additionally, consider condensing the related work to focus on key contributions that directly inform the development of any4, allowing for a more concise and impactful presentation of experimental results.

**Other Strengths And Weaknesses:**

### Strengths:

1. Originality – The any4 representation introduces a novel quantization approach that eliminates the need for preprocessing weights or activations, setting it apart from existing methods and encouraging further exploration in adaptive quantization.

2. Significance – With growing demand for efficient neural networks, any4 effectively reduces parameter size while maintaining high accuracy, making it valuable for both cloud and edge deployment of large language models.

3. Practical Implementation – The inclusion of tinygemm, a GPU-optimized matrix multiplication library, enhances the paper’s real-world applicability, achieving real speedup in GPUs.

4. Clear Evaluation – The paper rigorously benchmarks any4 against multiple numeric formats and quantization techniques, providing strong empirical evidence of its advantages.

### Weaknesses:

1. Limited Theoretical Foundation – While the experiments are thorough, a deeper theoretical analysis of any4’s effectiveness could strengthen its claims.

2. Calibration Dataset Concerns – Relying on a single curated calibration dataset may introduce biases or limit generalizability; a broader dataset selection could improve robustness.

3. Comparative Analysis Gaps – While the paper compares any4 with several quantization methods, expanding the discussion to include other optimization approaches, such as neural architecture search, could provide better context.

**Questions For Authors:**

1. What justifies the effectiveness of using only a single calibration sample? Would using more samples further improve performance? Additionally, how does the calibration cost of your method compare to other quantization techniques?

2. How does tinygemm interact with other acceleration methods, such as torch.compile, which can automatically optimize execution? Furthermore, when used in self-attention, how does its performance compare to FlashAttention (https://pytorch.org/blog/flashattention-3/)?

**Relation To Broader Scientific Literature:**

The paper introduces the any4 learned numeric representation, which improves traditional quantization techniques like int4 and fp4 without requiring preprocessing. It consistently outperforms or matches existing methods, using a single diverse sample for calibration instead of many. Results on group size and initialization enhance understanding of numeric stability. The paper suggests future work combining any4 with other techniques, contributing valuable insights to neural network efficiency and quantization research.

**Theoretical Claims:**

It does not explicitly detail any formal proofs or theoretical claims.

---

> ### Author Rebuttal · Authors · 2025-04-01
>
> We would like to thank the reviewer for their constructive review and comments, including that the approach was **“rigorously benchmark[ed]”**, the claims are **“supported by experimental results and methodological explanations”**, and that it’s implementation **“enhances the paper’s real-world applicability”**, as well as the important suggestions to improve the quality of the paper.
>
> Please find below our remarks regarding the reviewer’s requests and comments:
> - **Limited Theoretical Foundation / Formal Proof:** We would like to highlight that our approach is based on a theoretical derivation that we laid out in Section 4.1 of the paper. That mathematical derivation to minimize the reconstruction error of the output of each of the model’s linear layers, eventually lead to a modified k-Means clustering that guides us to apply clustering on the product of weights, mean activations, and group scaling factors (as described in Equation 14).
>   - To prove the claims of our formal proof, we have provided in this rebuttal Table A (in the response to Reviewer pDZ9), empirical results that prove that each term in Equation 14 was necessary to minimize the loss of the model.
> - **Calibration Dataset Concerns**:
>   - In Table A5 in the Appendix, we show that our single sample calibration set outperforms other datasets such as WikiText-2, C4, and The Pile.
>     - The results show that our approach of a single curated sample with sentences covering diverse topics (fiction, non-fiction, code, and math), as shown in Section A.2 in the Appendix could be sufficient to calibrate a quantization algorithm, in contrast to large datasets where each sample typically covers a single domain.
>   - Nevertheless, our approach can work by calibrating with arbitrary datasets with arbitrary number of samples.
> - **Calibration Cost:**
>   - Applying our single-sample calibration takes only 1–3 seconds, while applying a larger number of samples (e.g., 128) that is required by other algorithms could take 10 or 20 seconds.
>   - We have found that the total time to run our quantization algorithm on Llama2 7B is 10 minutes, which is similar to the time reported by other quantization algorithms: AWQ and GPTQ.
> - **Interaction with Other Acceleration Methods:** Overall, our quantization algorithm is orthogonal to other optimizations.
>   - **Torch.compile:** We have already integrated our INT4 implementation of our tinygemm library with torch.compile. We still haven’t yet integrated the ANY4 implementation within the library with torch.compile, but we believe it is straightforward.
>   - **FlashAttention:** FlashAttention in general (and FlashAttentionv3 in particular) only optimizes activation-activation matrix multiplication in the attention operation (i.e., Y = softmax(QK.T / sqrt(d) )V) while our work focuses on quantizing weight-activation matrix multiplication (for attention that would be Q=XWq, K=XWk, V=XWv, O=XWo, and we also quantize all the matrix multiplications in feed forward layers). Hence, from the perspective of end-to-end speedup of a model, our work is orthogonal to FlashAttention.
> - **Other**
>   - **Moving Results from Appendix to Main Body:** We agree with this proposal and plan to do that in the Camera Ready version if the paper is accepted.
>   - **Discussions on Other Optimization Approaches:** We can add discussions to the paper but we can mention here:
>     - Quantization in general is orthogonal to other optimization approaches like pruning and neural architecture search.
>     - We have focused in this paper on Post-Training Quantization (PTQ) that does not require retraining or fine-tuning of the model, while pruning requires extensive fine-tuning or continual pretraining, and NAS requires training from scratch or continual pretraining.
> - **Strengthening Evaluation:**
>   - **Ensuring Statistical Validation:** For the camera ready paper, we plan to re-run some of the text generation experiments with different seeds and report the average, and measure perplexity on different random subsets of the respective datasets and report their averages. However, we have noticed that in most papers on quantization, and on LLMs in general, measurements on perplexity as well as text generation are only provided for a default seed.
>   - **Consistent Experimental Conditions:** For perplexity, we have re-used the same evaluation script that is used in [GPTQ](https://github.com/IST-DASLab/gptq/blob/2d65066eeb06a5c9ff5184d8cebdf33662c67faf/llama.py#L206), that is also used in [AWQ](https://github.com/mit-han-lab/llm-awq/blob/aacd3b8923b080d58734001b3e7842c8ca3e6967/awq/entry.py#L300), and used the exact same seed to select the subsets of data. For downstream tasks, we have used LM Evaluation Harness and BigCode Evaluation Harness with the default settings provided in each library. We believe that our experimental conditions are consistent but we welcome further suggestions to improve consistency.

---

### Official Review · Reviewer_yZMB · 2025-03-13

**Overall Recommendation:** 3

**Summary:**

The paper presents any4, a new 4-bit numerical representation for post-training quantization of LLMs. The paper states that any4 does not require any additional pre-processing of weights or activations and for most LLMs, can find the optimal 4-bit representation with a single sample. The authors define the basis for the any4 format - using weight-only group quantization. They derive the rest of the math which forms the basis of their k-means based algorithm. This results in an LUT representation, each having 16 elements (for 4-bits). The authors further present tinygemm, a library that has efficient implementations for the different numerical formats (primarily matrix multiplies are achieved via efficient dequantization of weights to 16-bits and then multiplying with the activations). This is followed up by experimental results for different models (Llama 2/3, Mistral), showcasing both perplexity and downstream few-shot evaluation results.

---------

### Update after rebuttal

After reviewing the rebuttal and the responses to follow-up questions, I've decided to retain my score. There are several reasons:
- The difference between the LUT compression row-wise and group-wise quantization introduces a disparity in my mind. It seems like throwing away many of the benefits of group quantization to save memory with row-wise LUTs. It might have been nice to explore other forms of quantization like power of 2 scaling from MX formats to save on memory (uses only a scale factor with 8-bit scales).
- The RTN explanation still feels lacking. In the end, the authors do apply changes like group wise quantization and an LUT lookup table - for which I'd categorize this method as an LUT method.
- The overall novelty of the algorithm is limited. LUT compression has been explored in various forms before - and with upcoming fp4 formats in GPUs, having 4-bit codebooks is an expensive choice.

**Claims And Evidence:**

I think one thing that is unclear is the differentiation between the group quantization approach which the authors state they use for their method vs the computation of the LUT across an entire row (both in their derivation (equations 15-23), but also in their Figure 2). This is somewhat confusing. What is the benefit of doing group quantization and then creating only row-wise LUT? The end result is many scalars for the actual quantization and then very few actually representable values to represent them. Is this not creating a mismatch in the overall quantization process?

**Essential References Not Discussed:**

N/A

**Experimental Designs Or Analyses:**

Yes, the experimental analysis and presented ablations are sound.

**Methods And Evaluation Criteria:**

Yes, the presented evaluations are consistent with other literature in the field, and do fair comparisons against other methods.

**Other Comments Or Suggestions:**

1. Table 2 - for the numeric format column, this should talk about the format and not the method used
2. Lines 307-308, there is no figure 4.1 - please correct this to point to Figure 2 instead
3. Table A2 needs the correct highlighting for results of Mistral-8x7B model

**Other Strengths And Weaknesses:**

Addn. Strenghts:
- The authors present benchmarks on generative tasks like MBPP - which gives higher confidence in the presented method

Weaknesses:
- Despite using a scalar based 4-bit LUT quantization method, the authors do not compare their results with SqueezeLLM, which is the closest method to their paper.
- Broadly classifying their algorithm under the RTN umbrella is not correct. There are additional transformations going on to ensure high quality of their resulting models - Table 2 needs to fix this.
- While the overall method does show some benefits on certain tasks, the results are varied for different methods and different tasks. For example in table 2, sometimes their method is better, and sometimes its not. Same with Table A1.

**Questions For Authors:**

N/A

**Relation To Broader Scientific Literature:**

The paper presents a new 4-bit numerical representation, which is similar to previous efforts such as NF4 [1] and FP4 [2]. For the new method, the authors work with group quantization, which has been tackled in previous works such as GPTQ [3], and their benchmarks are commonly reported in quantization papers [3].

[1] NF4: https://arxiv.org/abs/2305.14314
[2] FP4: https://arxiv.org/abs/2310.16836
[3] GPTQ: https://arxiv.org/abs/2210.17323

**Theoretical Claims:**

While there are no theoertical claims, the authors present the full derivation of their method. I have checked the soundness of their math.

---

> ### Author Rebuttal · Authors · 2025-04-01
>
> We would like to thank the reviewer for their constructive review, including **“higher confidence in the presented method”** due to presenting results on generation and coding tasks, and thank the reviewer for checking the **“soundness of [the] math”** of the **“full derivation of [the] method”**, as well as for the important suggestions to improve the quality of the paper.
>
> Please find below our remarks:
> - **Benefit of Group Quantization with Row-wise LUT:** the reason we chose row-wise (look up table) LUT is to minimize the storage overhead.
>   - Each LUT consists of 2^n_bit FP16 values. So for 4-bits, each LUT consists of 16 FP16 values. Having such a LUT for each group of 128 values is a high overhead, while having it for each row (in Llama2 7B each row has 4096 values) the overhead is negligible.
>   - On the other hand, grouping requires 2 FP16 values (one for scale and one for offset) for each group. So we can afford having relatively small group sizes for grouping.
> - **Comparison with SqueezeLLM:** In the paper we haven’t compared with SqueezeLLM because SqueezeLLM keeps a portion (albeit small) of it’s weights in high precision (in their code they keep 10 rows of each weight matrix in high precision as well as 0.45% of outlier and sensitive values in the matrix). Nevertheless, we compare it here as requested by the reviewer in the Table below.
>   - Despite that any4 does not rely on storing rows or portions of its weights in high precision, its perplexity is competitive with SqueezeLLM.
>   - Moreover, we would like to highlight that storing rows or outlier/sensitive values in high precision is orthogonal to any4 and could be combined with it.
>   - The numbers reported below were copied from the SqueezeLLM paper. We tried to run the SqueezeLLM code to quantize Llama3 models, to compare with any4, but we hit into runtime errors in their code.
>
> | Model      | Quantization | WikiText-2 PPL↓ | C4 PPL↓ |
> |------------|--------------|-----------------|---------|
> | Llama2 7B  | SqueezeLLM   | 5.57            | 7.08    |
> |            | ANY          | 5.59            | 7.10    |
> | Llama2 13B | SqueezeLLM   | 4.96            | 6.54    |
> |            | ANY          | 4.97            | 6.55    |
> | Llama2 70B | SqueezeLLM   | 3.39            | 5.57    |
> |            | ANY          | 3.40            | 5.58    |
>
> **Table D:** *Comparison of any4 with SqueezeLLM*
>
> - **Any4 Under RTN Umbrella:** We categorized ANY4 under round-to-nearest (RTN) because in RTN given a list of possible values (whether those values are a LUT as in ANY4, or predefined as in INT4, FP4, or NF4), we round each weight to the nearest one.
>   - Any computation done in ANY4 algorithm is used to obtain the optimal values of the LUT, rather than modifying the values of the weights or activations. While quantization algorithms do modify the values of weights and/or activations:
>     - AWQ scales down the activations and scales up the weights.
>     - GPTQ quantizes each column in a weight matrix sequentially, and every time a column is rounded to the nearest number, the remaining unquantized weight values are modified to mitigate the reconstruction error of rounding previous columns.
> - **Accuracy on All Tasks:**
>   - **Perplexity is a Less Noisy Indicator then Downstream Tasks:** Accuracies on downstream tasks, especially generation tasks, could be noisy. EvalArena [D] studies the noisiness of such downstream tasks. E.g., HumanEval and MBPP appear in EvalArena with a significantly low signal to noise ratio, which explains why results in Table A1 may show ANY4 not always outperforming others on those specific tasks.. On the other hand, perplexity is a less noisy metric as it measures average loss on all tokens in each sample (rather than evaluating a final token or evaluating a pass or fail for a whole sample), and any4 does consistently well on perplexity in Table A1.
>   - **4-bit vs. 3-bit vs. 2-bit:** Table 1 compares perplexity on various bitwidths with quantization algorithms that do process weights and/or activations. While the results show QuIP tends to be better than our approach on lower bits (always better on 2-bits and sometimes better on 3-bits), our approach is always in the top for 4-bits.
>   - **Overall:** instances where any4 is not the top-performing method are a natural outcome of our extensive evaluation across a diverse set of models, generation types, model sizes, and tasks.
> - **Other:**
>   - **Typos and Formatting:** We thank the reviewer for pointing out the typos and issues in formatting. We have applied all the fixes.
>   - **Theoretical Claims:** Please see "Limited Theoretical Foundation / Formal Proof" in our response to Reviewer aGNw.
>
> **References**
> [D] EvalArena, https://crux-eval.github.io/eval-arena/

---

> > ### Comment · Reviewer_yZMB · 2025-04-08
> >
> > Thanks to the authors for all the reviews and responses. Based on this, I am leaning towards keeping my score as is (ie, leaning towards an acceptance).

---

### Official Review · Reviewer_pDZ9 · 2025-03-23

**Overall Recommendation:** 2

**Summary:**

The paper introduces a method for finding the optimal 4-bit quantization codebook for quantizing pre-trained language models. This is done by applying the LLoyd-Max algorithm to each *row* of the weight matrices of the model, thus finding an optimal (per-row) quantization look-up table. In order to find the optimal quantization values, the authors define the mean-squared error criteria not in terms of the weights themselves, but in terms of the outputs of the input-weight matrix multiplication, using a short hand-crafted input text sequence to achieve the calibration. They show that the obtained representation, called any4, performs favorably compared to quantizing to int4, fp4, and nf4 formats.

**Claims And Evidence:**

The main claim of the paper is that using the proposed any4 representation leads to better perplexity and task performance than quantizing to standard int4, fp4, and nf4 formats. The authors test their method on Llama 3 models of different sizes (1B, 3B, 8B, and 70B) shown in Figure 1 and  Table 1, as well as Llama 2 (7B, 13B, and 70B) in Table A1, and Mistral-7B and Mixtral-8x7B in Table A2. Overall, any4 does show improved perplexity, which tends to lead to better downstream task performance (but is not always the case). For these results, the authors fixed the group size (for the common scaling factors) to 128. The results do seem to indicate that the difference in performance is more pronounced for Llama 3 family of models, compared to Llama 2 and Mistral.

The authors also compare perplexity results (on WikiText-2) of their method vs. using post-training quantization techniques with integer formats in Table 2, showing it performs same/better for 4 bits and LLama 3 70B for 3 bits (the increase in perplexity for 3 and 2 bits is overall quite high for all techniques, so I am not completely sure about the utility of those particular results). It would be interesting to see PTQ results using NF4 (as this was the most competitive format otherwise), but I understand these results were taken from Huang et al., 2024 where integer formats were used.

**Essential References Not Discussed:**

I am not aware of any essential references that were not mentioned in the manuscript.

**Experimental Designs Or Analyses:**

/

**Methods And Evaluation Criteria:**

The authors evaluate their method using the perplexity metric on WikiText-2, C4, and Penn Treebank, as well as on downstream tasks within EleutherAI’s Harness and BigCode’s Harness. These datasets and tasks are in line with what is generally used in literature, and I believe provide a good evaluation basis for quantization approaches.

**Other Comments Or Suggestions:**

* Some links within the paper (to figures and tables) seem to be broken (for example Fig. 4.1, line 307).
* The authors might want to consider moving some of the derivation on pages 5 and 6 to the appendix to improve the flow of the main paper, but I leave this to the authors’ preference.
* It might be useful to specify for Eq. (6) what dequant function is (before it is expanded later in (12)).
* I believe it would be clearer to use the norm notation in the equations only for the vector equations (i.e. (10)), and not for the scalar equations such as (11).
* It might be useful to add a more detailed caption for Figure 2.
* I think the presentation of Table 2 could be made clearer (and “Numeric Format” column for 4-bits is wrong). For example, maybe consider removing “Numeric Format” columns altogether, and indicating next to the PTQ methods that they are associated with int format, while the last RTN row is associated with the any format.
* (Minor) There are two bolded values in Table 1, MBPP column, Llama 3.2 1B

**Other Strengths And Weaknesses:**

Strengths:
* The proposed method overall showcases better perplexity/task performance compared to the standard numerical formats.
* Good literature review — I believe the authors provided a good coverage of different quantization techniques.
* Providing an open-source GPU implementation of the method is a very welcome contribution.

Weaknesses:
* I am slightly concerned that the paper lacks novelty; the main contributions of the paper are using the standard Lloyd-Max procedure per row of weight matrices, as well as defining the problem in terms of the matrix multiplication outputs using a short hand-written calibration dataset. It doesn’t seem too surprising that finding an optimal quantization codebook would lead to better performance compared to pre-defined quantization formats; in addition, the improvement does not seem to be as significant for models other than Llama 3 family.

**Questions For Authors:**

The following questions are more of minor/clarifying nature:
* As the authors compare their technique with a few other PTQ techniques, I was wondering if they could provide a comparison  (could just be a comment/approximate) in terms of the cost of their k-means technique vs. running a PTQ algorithm?
* Why is group size=128 used for the main results? (I also thought it might be useful to add group size=32 results, as this is the standard group size used for e.g. MXFP)
* Would be interesting to know how much better is it to optimize outputs on a calibration dataset vs. optimizing the weights directly?
* The results for any4 seem to be stronger for Llama 3 compared to other model families explored; do the authors have any comments/explanation for this?

**Relation To Broader Scientific Literature:**

I believe the authors did a fine job of relating their work to the prior literature; they use the well-known Lloyd-Max algorithm to find optimal quantization codebooks for each row of every weight matrix within the model, and compare this to quantizing to standard numerical data types, as well as the commonly used post-training quantization techniques.

**Theoretical Claims:**

The main theoretical derivation in the paper is for the k-means/Lloyd-Max algorithm for finding the optimal quantization codebook given the set of weight elements. The algorithm is fairly standard, and I believe its usage within this work is appropriate. I believe the final algorithm is correct, I just wanted to ask a clarifying question:
* The authors claim that Eq. (17) follows from (14); this wasn’t quite clear to me, as (14) involves the summation across all row elements, while in (17) we are choosing the closest quantized value for each element in the row independently. Couldn’t the optimal weights for (14) differ from (17), i.e., (17) is an approximation but not the exact solution to (14)?

---

> ### Author Rebuttal · Authors · 2025-04-01
>
> We thank the reviewer for their detailed and constructive feedback, which will help improve the paper.
>
> Please find below our remarks:
> - **Optimizing Weights Directly:** Please find the results in Table A below. First row shows the results of optimizing weights directly. The other 2 rows show the results of using the 2 additional terms of Equation 14 in our paper, i.e., multiplying with activations and scales. These results confirm that our derivation (Eq. 14) is essential for optimal performance.
>
> | | Term to Minimize in Equation 14 | WikiText-2 | C4 | PTB  | CodeParrot |
> |--|--|--|--|--|--|
> | Optimizing Weights Only | $( w_{S_{i,j}} - w_{Q_{i,j}} )$ | 6.680 | 9.619 | 11.186 | 2.751 |
> | Optimizing Weights * Activations | $( w_{S_{i,j}} x_{j} - w_{Q_{i,j}} x_{j} )$ | 6.496 | 9.375 | 11.055 | 2.675 |
> | Optimizing Weights * Activations * Group Scales [Ours] | $(\alpha_{i,j} w_{S_{i,j}} x_{j} - \alpha_{i,j} w_{Q_{i,j}} x_{j} )$ | 6.487 | 9.366 | 11.034 | 2.680 |
>
> **Table A:** *PPL Results on Any4 Quantization on Llama3.2 1B (Lower is Better)*
>
> - **Lack of Novelty:** Please see our response to Reviewer U2Ld
> - **Significant Improvement for Llama3:** Quantizing Llama 3 is known to be more challenging than Llama 1 and 2, often resulting in larger accuracy drops [A]. This is likely due to its larger pretraining dataset (8T tokens vs. 2T for Llama 2 [B, C]), which makes it better trained—and therefore harder to compress without loss. In contrast, older models are relatively undertrained and easier to quantize, so the impact of advanced quantization methods appears smaller. Thus, the stronger results on Llama 3 highlight the effectiveness of our approach, especially on newer, more robust models where quantization is harder and matters more.
> - **PTQ Results using NF4:** We provide in Table B the results of combining AWQ with NF4 as well as results of combining AWQ with ANY4. These results show that combining preprocessing-based methods like AWQ with our numeric format leads to further improvements.
>
> | Model | Quantization Algorithm | Numeric Format | WikiText-2 PPL↓ |
> |--|--|--|--|
> | Llama3 8B | | FP16  | 6.14 |
> | | RTN | INT4 | 6.87 |
> | | RTN | NF4 | 6.63 |
> | | RTN | ANY4 | 6.51 |
> | | AWQ | INT4 | 6.53 |
> | | AWQ | NF4 | 6.51  |
> | | AWQ | ANY4 | 6.38 |
>
> **Table B:** *Combining AWQ with Different Numeric Formats*
>
> - **Algorithm Time Comparison:** We have measured the time to quantize a Llama2 7B and found it to be approximately 10 minutes, which is similar to the time we have measured to run both AWQ and GPTQ. Therefore, we conclude that quantization times of different approaches are similar.
> - **Different Group Sizes:**
>   - We used group size 128 as it is the default in many quantization papers (e.g., AWQ, GPTQ).
>   - In the Appendix we have provided results for group sizes 64, 128, 256, 512, 1024 for Llama3.2 1B.
>   - As requested by the reviewer, we also provide here results for group size 32, as well as the other group sizes we already had in the Appendix
>   - Please note the bitsandbytes library that we use for FP4 and NF4 quantization doesn't support group size 32.
> | | 32 | 64 | 128 | 256 | 512 | 1024 |
> |--|-|-|-|-|-|-|
> | FP4  | N/A   | 16.19 | 17.11 | 18.12 | 20.43 | 2.3E6 |
> | NF4  | N/A   | 14.27 | 14.63 | 14.98 | 15.38 | 7.8E5 |
> | ANY4 | 13.54 | 13.75 | 13.95 | 14.09 | 14.24 | 14.34 |
>
> **Table C:** *Llama3.2 1B Perplexity on C4 (Lower is Better)*
>
> - Other:
>   - **Formatting and Typos:** We thank the reviewer for the detailed comments and we will apply them to the camera ready paper.
>   - **Equation 17 following Equation 14:** We thank the reviewer for highlighting this important point. While Eq. (14) defines a global row-wise objective, Eq. (17) corresponds to a local minimization step within a K-Means-style alternating optimization procedure.
>     - Specifically, Eq. (17) performs the E-step, assigning each weight $w_{s_{i,j}}$ to its nearest codebook value in $Q_i$, treating activations $x_j$ as constants. This simplifies to a local nearest-neighbor assignment. The M-step (Eqs. (19)–(20)) then updates $Q_i$ by minimizing the total reconstruction error across the row.
>     - Thus, while Eq. (17) does not solve the global objective in Eq. (14) directly, it is part of an iterative process that does. We will revise the text to clarify this decomposition and explicitly note that Eq. (17) performs local minimization within the broader alternating optimization scheme.
>   - **Downstream Task Performance:** Please check "Perplexity is a Less Noisy Indicator then Downstream Tasks" in our response to Reviewer yZMB.
>
> **References**
> [A] “How Good Are Low-bit Quantized LLaMA3 Models? An Empirical Study”, https://arxiv.org/abs/2404.14047v1, April 2024
>
> [B] “Scaling Laws for Floating Point Quantization Training”, https://arxiv.org/abs/2501.02423, January 2025
>
> [C] “The Llama 3 Herd of Models”, https://arxiv.org/abs/2407.21783, July 2024

---

> > ### Comment · Reviewer_pDZ9 · 2025-04-08
> >
> > I'd like to thank the authors for the detailed and informative response, including the additional results included. While I still have some reservations regarding the overall impact of the work, I do believe the evaluation is thorough and it effectively showcases the benefits of the proposed method, which the added results further confirm. Although I am inclined to keep my initial overall score, I am leaning towards an overall positive assessment of the paper.

---

### Decision · Program_Chairs · 2025-05-01

**Decision:**

Accept (poster)

**Comment:**

Despite low scores, reviewers all point to the value of this contribution, and it is clear that it is timely and sufficiently novel for ICML Low-precision computation is relevant across a wide range of ML workloads, and hence is of interest to a large subset of the ICML audience. As noted in review, the implications of this work will change with newer hardware architectures, but in my view, this strengthens the case for timely publication.